# OsCAF1, a CRM Domain Containing Protein, Influences Chloroplast Development

**DOI:** 10.3390/ijms20184386

**Published:** 2019-09-06

**Authors:** Qiang Zhang, Lan Shen, Zhongwei Wang, Guanglian Hu, Deyong Ren, Jiang Hu, Li Zhu, Zhenyu Gao, Guangheng Zhang, Longbiao Guo, Dali Zeng, Qian Qian

**Affiliations:** 1State Key Laboratory of Rice Biology/China National Rice Research Institute, Chinese Academy of Agricultural Sciences, Hangzhou 310006, China; 2Biotechnology Research Center, Chongqing Academy of Agricultural Sciences, Chongqing 401329, China

**Keywords:** chloroplast RNA splicing and ribosome maturation (CRM) domain, intron splicing, chloroplast development, rice

## Abstract

The chloroplast RNA splicing and ribosome maturation (CRM) domain proteins are involved in the splicing of chloroplast gene introns. Numerous CRM domain proteins have been reported to play key roles in chloroplast development in several plant species. However, the functions of CRM domain proteins in chloroplast development in rice remain poorly understood. In the study, we generated *oscaf1* albino mutants, which eventually died at the seedling stage, through the editing of *OsCAF1* with two CRM domains using CRISPR/Cas9 technology. The mesophyll cells in *oscaf1* mutant had decreased chloroplast numbers and damaged chloroplast structures. OsCAF1 was located in the chloroplast, and transcripts revealed high levels in green tissues. In addition, the OsCAF1 promoted the splicing of group IIA and group IIB introns, unlike orthologous proteins of AtCAF1 and ZmCAF1, which only affected the splicing of subgroup IIB introns. We also observed that the C-terminal of OsCAF1 interacts with OsCRS2, and OsCAF1–OsCRS2 complex may participate in the splicing of group IIA and group IIB introns in rice chloroplasts. OsCAF1 regulates chloroplast development by influencing the splicing of group II introns.

## 1. Introduction

Chloroplasts are important organelles in plants. A series of metabolic processes occur in chloroplasts, including photosynthesis and anabolism of compounds such as tetrapyrroles, terpenoids, lipids, amino acids, and hormones [1]. Chloroplasts are considered semi-autonomous organelles because their development is not only influenced by their own genetic material but also by fine regulation by nuclear-encoded genes [2,3]. Studies have shown that by plastid-encoded polymerases (PEPs) and nucleus-encoded polymerases (NEPs) influence the development of chloroplasts [4,5,6]. Chloroplast genes encode approximately 100 proteins, some of which have one or more introns that cannot be self-spliced, and the primary RNA transcription of such chloroplast genes require splicing by ribozymes potentially via chemical steps similar to spliceosome-mediated splicing in the nucleus [7,8,9]. In plants, based on the primary sequences, predicted structures, and splicing mechanisms, the introns of the chloroplast are mainly classified into two categories, including group I and group II [10]. Group II introns are mainly divided further into two subgroups, including subgroup IIA introns and subgroup IIB introns [11]. *Arabidopsis thaliana*, maize, and rice chloroplast genomes all have only 1 group I intron, and 20, 17, and 17 group II introns, respectively [8]. In plant chloroplasts, such introns have lost the capacity to self-splice in vivo and require nuclear gene-encoded proteins as co-factors to participate in splicing [8,12].

Numerous studies have shown that nuclear-encoded pentatricopeptide repeat (PPR) proteins participate in chloroplast RNA editing and splicing, which are critical for chloroplast development and function [8,13]. Currently, the large PPR protein families, including AtOTP51, ZmPPR5, OsPPR6, and OsPPR1, are considered to participate in different chloroplast group II intron splicing activities [14,15,16,17,18]. Previous studies have shown that the disruption of the normal functions of such PPR proteins could lead to abnormal chloroplast development, which would lead to albino seedlings or death of plants [14,15,16,17,18]. In addition to the PPR proteins mentioned above, chloroplast RNA splicing and ribosome maturation (CRM) domain proteins participate in the splicing of chloroplast gene introns [8]. 

In plants, the splicing of chloroplast introns requires splicing factors encoded by nuclear genes and the abnormal splicing of chloroplast intron will affect the development of chloroplast [14,15,16]. Several proteins with CRM domains have been identified as splicing co-factors, including CFM2, CFM3, CRS1, CAF1, and CAF2 [8]. Both ZmCFM2 and AtCFM2 have four CRM domains, and they participate in the splicing of *ndhA* and *ycf3-1* subgroup IIB and group I introns. In *A. thaliana,* AtCFM2 potentially promotes the splicing of *clpP* introns [19]. CFM3 and CRS1 contain three CRM domains, and AtCRS1 and ZmCRS1 have been associated with the splicing of the *atpF* intron, which belongs to subgroup IIA introns [20,21]. In addition, CFM3 has been reported to be dual-localized in chloroplasts and mitochondria, and CFM3a is required for the splicing of group II introns, including *ndhB*, *rpl16*, *rps16*, *petD*, and *petB* introns in chloroplasts [19]. The CAF1 and CAF2 contain two CRM domains, which are required for the splicing of group IIB introns, including *ndhA*, *ndhB*, *petB*, *petD*, *rpl16*, *rps16*, *trnG*, and *ycf3-1* in maize and *A. thaliana* [11,21]. In *A. thaliana,* AtCAF2 potentially promotes the splicing of *rpoC1* and *ClpP* introns that are absent in the maize chloroplast genome [21]. In addition, CAF1 and CAF2 could interact with CRS2, forming the CRS2–CAF1 and CRS2–CAF2 complexes, which participate in the splicing of chloroplast group II introns and the regulation of chloroplast development in maize [22]. 

Previous studies have shown that there are 14 proteins that contain one or more CRM domains in rice [8]. To date, researchers have only studied the functions of OsCRS1 and OsCFM3, which contain three CRM domains. The studies revealed that *oscrs1* mutants exhibited albino leaf phenotypes in rice [23]. In addition, OsCRS1 has been reported to participate in the splicing of *ndhA*, *ndhB*, *petD*, *ycf3-1*, and *trnL* introns and it could influence the expression of PEP-dependent genes such as *psaA* and *psbA* [23]. Furthermore, *oscfm3* T-DNA insertion mutants exhibited albino seedling phenotypes; however, the OsCFM3 participates in the intron splicing of *ndhB*, *petD*, *rps16,* and *rpl16* in rice [19]. The functions of other CRM domain proteins have hardly been reported in rice. Previous studies have shown that homologous *CAF1* plays important roles in chloroplast development in *A. thaliana* and *Zea mays* L. In rice, the function of *OsCAF1* remains unknown and investigating its function would enhance our understanding of mechanisms of chloroplast development.

In this study, we investigated the function of OsCAF1 in rice chloroplast development, and *oscaf1* mutants were obtained using the CRISPR-Cas9 system. The results of our study revealed that *oscaf1* mutants exhibited albino seedlings phenotype along with a damaged chloroplast structure. In rice, chloroplast development is regulated by the expression of PEP-dependent genes, which could be affected by *OsCAF1*. In addition, we observed that the leaves of *oscaf1* mutants accumulated higher hydrogen peroxide (H_2_O_2_) contents. In addition, OsCAF1 could interact with the OsCRS2 via C-terminal, and then form an OsCAF1–OsCRS2 complex that regulates the splicing of chloroplast gene introns. Chloroplast RNA splicing analysis showed that OsCAF1 influenced the splicing of both chloroplast subgroup IIA and subgroup IIB introns, which is different from the orthologous proteins of AtCAF1 and ZmCAF1, which both influence the splicing of chloroplast subgroup IIB introns. The results of the present study reveal the functions of OsCAF1 in the regulation of rice chloroplast development.

## 2. Results

### 2.1. Knockout of OsCAF1 Produced the Albino Seedling Phenotype

According to a previous study, the ortholog of *ZmCAF1* in rice was *Os01g0495900*, which was named *OsCAF1*, and the function of OsCAF1 has not been determined to date [8]. We observed that the cDNA of *OsCAF1* had a 2106-bp nucleotide and encoded 701 amino acids (aa). To determine the function of OsCAF1 in rice chloroplast development, *oscaf1* mutants were generated using the CRISPR/Cas9 system (Figure 1A,B), and 26 transgenic plants were obtained in the T_0_ generation. Sequencing and phenotypic analysis revealed that *oscaf1* homozygous mutants exhibited the albino phenotype in the rice seeding stage (Figure 1C,D), and then died after the three leaves stage. In both of #3 and #4 mutants, a premature stop codon was created by a frame shift in the coding region of *OsCAF1*. However, the *OsCAF1/oscaf1* heterozygous exhibited green leaves (Figure 1C,D), which exhibited normal growth and phenotypes under chamber conditions. Because of *oscaf1/oscaf1* homozygous lethal mutants, the heterozygous lines were used to generate the T_1_ generation. The different homozygous mutations in the allele’s of *OsCAF1* in T_1_ generation were obtained from #1 and #2 plants and also exhibited albino seedling phenotypes. These results indicated that *OsCAF1* mutations caused the albino seedling phenotype. To study the function of OsCAF1, *oscaf1* mutants from T_1_ generation of #1 plant were selected for further analysis. 

### 2.2. The oscaf1 Mutant Exhibited Defects in Chloroplast Development

The *oscaf1* mutant exhibited an albino seedling phenotype and eventually died after the three leaves stage (Figure 2A). Compared with the wild type (WT), photosynthetic pigment concentrations showed that chlorophyll a (Chla), chlorophyll b (Chlb), and carotenoid (Car) contents decreased significantly in *oscaf1* mutants (Figure 2B). According to the results of chlorophyll fluorescence measurements, the Fv/Fm values of *oscaf1* mutants were low (Figure 2C). The results suggested that *oscaf1* mutants had impaired chloroplast development and photosynthesis. Subsequently, we analyzed the expression levels of chloroplast development and photosynthesis-related genes in WT and *oscaf1* mutants using qRT-PCR. The results showed that compared with the WT, the transcript levels of *HZMA1*, *PORA*, *CAB1*, *psaA*, *rbcL*, *CHLI*, *CHLH*, *ATPa*, *psbA*, *TaX2*, *ATPB*, and *ATPE* in *oscaf1* mutants decreased, while the relative expression levels of *YGLI*, *HSA1*, *HSA2*, *RPOB*, *RPOC1*, and *RPOC2* increased (Figure 2D). Particularly, in *oscaf1* mutants, the expression levels of *HZMA1*, *PORA*, *CAB1*, *psaA*, *rbcL*, *CHLH,* and *psbA* decreased significantly, and the expression levels of *YGLI*, *HSA1*, *RPOB*, *RPOC1,* and *RPOC2* increased markedly in *oscaf1* mutants (Figure 2D). Overall, the results suggested that OsCAF1 influences chloroplast development and photosynthesis in rice.

To investigate chloroplast development in *oscaf1* mutants further, we observed the ultrastructure of chloroplasts at the three leaves stages of WT and *oscaf1* mutants using transmission electron microscopy (TEM). We observed that the chloroplast number in the mesophyll cells of the *oscaf1* mutant was less than in WT (Figure 3B). In addition, the chloroplasts in WT mesophyll cells were well developed, with normal-looking and distinctly stacked grana and thylakoids (Figure 3A,C). Conversely, the *oscaf1* mutant had abnormal chloroplast architecture along with abnormally structured thylakoid and abnormally stacked grana (Figure 3B,D). The results suggested that OsCAF1 plays a key role in early chloroplast development in rice.

### 2.3. Increased Reactive Oxygen Species (ROS) Contents and Reduced ROS-scavenging Gene Expression Levels in oscaf1 Mutant

In plants, the overproduction of ROS might cause cell death in leaves [24]. The *oscaf1* mutants died after the three leaves stage; therefore, we tested the ROS levels in leaves of the WTs and *oscaf1* mutants. The *oscaf1* mutant leaves displayed a more intense brown color following 3,3-diaminobenzidine (DAB) staining, indicating that the levels of H_2_O_2_ in *oscaf1* mutant leaves were higher than in WT leaves (Figure 4A). In addition, following nitroblue tetrazolium (NBT) staining, *oscaf1* mutant leaves had larger blue leaf areas than WT leaves. The results suggested that oscaf1 mutant leaves generate more O^2−^ than WT (Figure 4B). The results of the experiments indicated that oscaf1 leaves accumulated high ROS levels. Moreover, we measured the H_2_O_2_ contents in WT and *oscaf1* mutant leaves. The results indicated that the H_2_O_2_ content in the mutant was significantly increased compared to that in WT (Figure 4C). This result was consistent with DAB staining. Meanwhile, the ROS-scavenging gene expression levels were investigated in WT and *oscaf1* mutants. According to the qRT-PCR results, the expression levels of *APX1*, *APX2*, *SODA1*, *SODB*, *CatA*, and *CatC* were significantly decreased in *oscaf1* mutants compared to WT (Figure 4B). The ROS-scavenging genes reduced expression in *oscaf1* might have had feedback from lack of OsCAF1. Overall, a decrease in ROS-scavenging gene expression levels would impair the ROS detoxification system, eventually leading to the accumulation of ROS, such as H_2_O_2_ in *oscaf1* mutant leaves.

### 2.4. OsCAF1 Involved in Splicing of Multiple Chloroplast Group II Introns

Previous studies have shown that *ZmCAF1* and *AtCAF1* participate in intron splicing in the chloroplast [21]. To test whether OsCAF1 was involved in the RNA splicing of chloroplast genes, we amplified the chloroplast genes that contained at least one intron in WT and *oscaf1* mutants using the RT-PCR method. We observed that the introns of chloroplast *atpF*, *rpl2*, and *rps12* could not be spliced, whereas the intron splicing efficiency of *ndhA*, *ndhB*, and *ycf3* decreased in group II introns in *oscaf1* mutants (Figure 5). In *oscaf1*, the splicing efficiency of group I introns (*trnL*) was normal. The results indicated that OsCAF1 might participate in the splicing of multiple chloroplast group II introns.

### 2.5. Subcellular Localization and Expression Patterns of OsCAF1

OsCAF1 was localized in the chloroplasts, which was predicted using the TargetP website (http://www.cbs.dtu.dk/services/TargetP/). To determine the actual subcellular localization of OsCAF1, we fused the OsCAF1 full-length cDNA with GFP driven by the 2× CaMV 35S promoter and transiently transfected in rice protoplasts. The results showed that the green fluorescent signals of OsCAF1–GFP co-localized with chloroplast autofluorescence in transformed rice protoplasts (Figure 6A). The assays indicated that OsCAF1 was localized in the rice chloroplast.

The qRT-PCR assays were used to investigate the expression patterns of OsCAF1 in different tissues at the three leaves stage. According to our results, OsCAF1 was highly expressed in green tissue, particularly in the leaves, suggesting that the OsCAF1 mainly functions in green tissue (Figure 6B).

### 2.6. OsCAF1 through C-terminal Interacts with OsCRS2 

In maize, CRS2 could interact with CAF1 and form a CRS2–CAF1 complex in chloroplasts [22]. Therefore, we investigated whether OsCAF1 interacted with OsCRS2 and formed an OsCRS2–OsCAF1 complex in rice. We obtained rice *OsCRS2* (*Os01g0132800*) by homologous alignment with *ZmCRS2* and observed that *OsCRS2* was also expressed highly in green tissue (Appendix A). The results of the yeast two-hybrid (Y2H) experiment revealed that OsCAF1 interacted with OsCRS2 and formed an OsCRS2–OsCAF1 complex in rice (Figure 7A). To determine which OsCAF1 section was essential for interactions with OsCRS2, we first generated three truncated forms of OsCAF1, including OsCAF1-N, OsCAF1-M, which contains two CRM domains, and OsCAF1-C (Figure 7B). The results showed that only OsCAF1-C could interact with OsCRS2, while OsCAF1-N and OsCAF1-M failed to interact with OsCRS2 (Figure 7C). The result indicated that the C-terminal of OsCAF1 was essential for interactions with OsCRS2. To further explore the function of the of OsCAF1 C-terminal, we analyzed the alignment of the C-terminals of OsCAF1 residues 387–701 aa in *Oryza sativa* with different species, including *Z. mays*, *A. thaliana*, *Panicum hallii*, *Brachypodium distachyon*, *Sorghum bicolor*, *Setaria italica*, *Triticum aestivum*, *Elaeis guineensis*, and *Phoenix dactylifera*. The sequence alignment results showed that the OsCAF1 C-terminal residues of 585–598 aa and 685–701 aa were highly conserved (Appendix A). To test the functional regions of the OsCAF1 in C-terminal, we divided the C-terminal containing OsCAF1 residues C1 (387–584 aa), C2 (585–598 aa), C3 (599–684 aa), and C4 (685–701 aa) (Figure 7B) for further study. The results of the Y2H experiment showed that OsCAF1-C1 could bind OsCRS2 (Figure 7C). The results indicated that the C1-terminal of OsCAF1 was required for interactions with OsCRS2 in rice; however, the two highly conserved regions of the C-terminal of OsCAF1 could not interact with OsCRS2.

The *oscaf1* mutant exhibited albinism and survived for only approximately three weeks. The chloroplast numbers in *oscaf1* mutants decreased, accompanied by abnormal chloroplast development (Appendix A) and damaged thylakoid membranes compared with the WT, which led to a decrease in chlorophyll contents in *oscaf1* mutants. Similar phenotypes of CRM domain proteins have been reported in rice, such as *oscfm3* and *oscrs1* [19,23]. OsCAF1 might play an important role in chloroplast development. Among the altered chloroplast and photosynthesis-related gene expressions in *oscaf1* mutants, PEP-dependent genes, including *PsaA*, *PsbA*, and *RbcL*, and encoding photosynthesis-associated proteins significantly decreased in *oscaf1* mutants, although NEP-dependent genes, *RpoC1* and *RpoC2*, markedly increased in *oscaf1* mutants. In addition, the expression levels of *AtpB* and *AtpE*, transcribed by both NEP and PEP, significantly decreased in *oscaf1* mutants. Therefore, OsCAF1 might regulate chloroplast development by influencing PEP activities.

The ROS contents such as H_2_O_2_ increased in the three leaves stage leaves of *oscaf1* mutants, compared with the WT. In addition, the expression levels of ROS-scavenging related genes decreased in *oscaf1* mutant leaves. Considering the results of a previous study [19], the functional obstruction of OsCAF1 could lead to abnormal chloroplast development, ROS over-accumulation in leaves, and eventually the death of rice seedlings after the three leaves stage.

Numerous leaf-color associated mutants have been identified and cloned in different plants [25,26,27]. Abnormal splicing of chloroplast gene introns could cause chlorophyll deficiency, which then leads to visible losses in green color in leaves or death of plants [25,26,27]. In rice, *wsl4* exhibit bleached appearances before the four leaves stage, and after the five leaves stage, all leaves turned to a normal color similar to WT under normal growth environments. The bleached appearance phenomenon in *wsl4* was caused by defective intron splicing of chloroplast genes, including *atpF*, *rpl2*, *ndhA*, and *rps12* [26]. The results of such studies indicate that the abnormal splicing of the four introns does not cause lethal phenotypes at seedling stages. In the present study, the splicing of *ndhB* and *ycf3* introns was also affected. The ndhB protein was an NADPH dehydrogenase and bound to the thylakoid membrane, influenced NDH activity, and regulated cyclic electron transport around photosystem I (PS I) and respiratory electron transport [28]. In tobacco, the *△ndhB* mutant exhibited abnormal intron splicing of only chloroplast *ndhB*, and the *△ndhB* mutant could grow normally under normal conditions [28]. In addition, both *crr2-1* and *crr2-2*, with impaired intron splicing of *ndhB* in *Arabidopsis*, could grow normally under normal conditions [29]. In the present study, NDH activity or electron transfer efficiency of PS I may be affected in *oscaf1* mutants; however, the impairing of the *ndhB* intron was not the key factor influencing seedling death after the three leaves stage. Another intron splicing impaired gene, *ycf3*, the chloroplast-encoded Ycf3 protein, was responsible for the stability of the PS I protein complex [16]. Previous studies have shown that numerous mutants, such as *otp51, osppr6,* and *tha8*, have albino seedling lethal phenotypes similar to the one in the *oscaf1* mutant, and all the mutants have similar genetic defects including reduced intron splicing efficiency of *ycf3,* similar to the results in the present study [14,16,27]. Therefore, the Ycf3 protein could be required for early chloroplast development. In *oscaf1* mutants, the reduced splicing efficiency of *ycf3* was responsible for mutant deaths after the three leaves stages.

In *Arabidopsis*, AtCAF1 is required for the splicing of group IIB introns, including *ndhA*, *petD*, *rpl16*, *rps16*, *trnG*, *ycf3*, *rpoC1*, and *ClpP* [21]. ZmCAF1 in maize is also required for the splicing of group IIB introns, almost similar to *A. thaliana*, excluding *rpoC1* and *ClpP* [11]. Previous results have shown that CAF1 is not required for group IIA intron splicing in maize and *A. thaliana* [11,21]. Notably, our results indicated that the efficiency of intron splicing of group IIA introns, including *atpF, rpl2*, and *rps12*, and group IIB introns, including *ndhB*, *ycf3*, and *ndhA*, were affected. Particularly, the group IIA introns were not spliced in *oscaf1* mutants completely. The results indicated that OsCAF1 could play a key role in the regulation of intron splicing in group IIA and group IIB introns in rice, which was different from maize and *A. thaliana*. Previous study showed that OsCRS1 not only participated in the splicing of *atpF* introns but also promoted the splicing of other introns, including *trnL, rpl2, ndhA, ndhB, petD*, and *ycf3* [22]. Therefore, the functions of OsCRS1 and OsCAF1 could partially overlap but are not redundant in rice. In addition, the intron splicing of subgroup IIB introns, including *petD, rpl16, rps16*, and *trG* does not require OsCAF1 in rice, while the splicing of the subgroup IIB introns requires ZmCAF1 and AtCAF1 in maize and *A. thaliana* [11,21]. Moreover, in rice, there are three additional proteins containing two CRM domains, and the functions of the proteins may be different from OsCAF1. Intron splicing of subgroup IIA and subgroup IIB introns was regulated by OsCAF1, which, in turn, influenced chloroplast development in rice.

Nuclear-encoded protein ZmCRS2 could interact with ZmCAF1 to form the CRS2–CAF1 complex, which regulates the splicing of group IIB introns in maize chloroplasts [22]. In addition, our study demonstrated that OsCAF1 could interact with OsCRS2 through the C-terminal region, suggesting that the interaction mechanism of CAF1 and CRS2 might be conserved in maize and rice. The homologous alignment analysis results of the C-terminal region of CAF1 among various plants suggested high conservation at C-terminal positions 585–598 aa and 685–701 aa. However, the results of our study showed that the C1 region of OsCAF1, 387–584 aa, was essential for interactions with OsCRS2 rather than the conserved regions of the C2 and C4 regions. The C2 and C4 conserved regions of OsCAF1 could interact with other proteins, which requires further investigations. In addition, the *OsCRS2* expression levels decreased significantly in *oscaf1* (Appendix A), suggesting the mutation of OsCAF1 also influenced *OsCRS2* expression. Based on the above findings, the C1 region of OsCAF1 interacts with OsCRS2, and the generated complex, OsCRS2–OsCAF1, could participate in the splicing of subgroup IIA and subgroup IIB introns in rice chloroplast.

OsCAF1, which has two CRM domains, is an RNA intron-splicing factor in rice. *OsCAF1* influences the expression of chloroplast-associated genes, affects the accumulation of H_2_O_2_ in rice leaves, and plays a key role in early chloroplast development in rice. We demonstrated that OsCAF1 is potentially involved in the splicing of both group IIA and group IIB introns of chloroplast transcripts in rice. In addition, the C1-terminal region of OsCAF1 could interact with OsCRS2 and the OsCRS2–OsCAF1 complex could participate in the splicing of group II introns. 

## 3. Materials and Methods

### 3.1. Plant Materials and Growth Conditions

The *oscaf1* mutants were obtained using the CRISPR-Cas9 gene editing system in the *japonica* rice variety, Nipponbare, which was used as the WT. For chlorophyll content measurement and genetic material and RNA extraction, the WT and *oscaf1* plants were grown in growth chambers under a 16 h light and 8 h of a dark cycle at a constant temperature of 30 °C.

### 3.2. Knockout of OsCAF1 Using the CRISPR/Cas9 System

The rice genome editing strategy was based on the CRISPR/Cas9 system according to a previous study [30]. Two gRNA target sequences (AAGCCCAGTACCCCATCTCACGG, CCGAGGTACCAAGCGGCGTCCAG) were designed from exon 1 to construct the intermediate vector (Figure 1B) namely SK-gRNA- g*^OsCAF1-g1^* and SK-gRNA- g*^OsCAF1-g2^*. The binary expression vector transferred into *Agrobacterium tumefaciens* strain EHA105 was constructed using the isocaudamer ligation method; the intermediate vectors were digested with *Kpn* I/*Xho* I, *Sal* I/*Bgl* II, and then assembled into the pC1300-Cas9 binary vector (digested with *Kpn* I/*BamH* I). The schematic for the development of the plant expression vector is illustrated in Figure 1A,B. The intermediate vector SK-gRNA and the expression vector pC1300-Cas9 were provided by Kejian Wang and were kept in our laboratory. All of the primers are listed in Appendix A.

### 3.3. Transmission Electron Microscopy (TEM)

The WT and *oscaf1* seedlings were grown in growth chambers, as above. The transmission electron microscopy (TEM) analysis was performed according to a previous study [23]. Briefly, the third leaves from WT and *oscaf1* were collected and cut into approximately 0.5 cm^2^ pieces. Leaf samples were fixed using 2.5% glutaraldehyde and 1% OsO_4_, dehydrated using an ethanol series, and embedded in resin. The samples were stained again with uranyl acetate and alkaline lead citrate and finally observed under a HitachiH-7500 TEM (Tokyo, Japan).

### 3.4. RNA Extraction and Quantitative Real-time PCR (qRT-PCR) Assay

Total rice RNA from leaves of *oscaf1* and WT plants was extracted using an RNA extraction Kit (TaKaRa, Japan). First-strand cDNA was synthesized using a ReverTra Ace qPCR RT Kit (TOYOBO, Japan). The qRT-PCR was conducted using a SYBR green real-time PCR master mix (TOYOBO, Japan) on a Bio-Rad CFX96 system according to the manufacturer’s instructions. The qRT-PCR procedure was as follows: 5 min at 95 °C followed by 40 cycles of 95 °C for 10 s and 58 °C for 1 min. *OsActin1* were used as internal controls. The fluorescence data were analysis by Lin-RegPCR program [31,32] to calculate primer efficiency and obtain Ct values. The genes relative expression levels were calculated according to previous study [33]. All qRT-PCR primers are listed in Appendix A. For date statistical significance was analyzed using ANOVA with Tukey post hoc pairwise comparisons. * and ** indicate *p* < 0.05 and *p* < 0.01, respectively. 

### 3.5. Chloroplast RNA Splicing Analysis

The cDNA of WT and *oscaf1* plants seeding leaves were obtained according to the procedures above. The RT-PCR procedure was as follows: 95 °C for 5 min, followed by 32 cycles of 95 °C for 30 s, 60 °C for 30 s, 72 °C for 1 min, and a final elongation step at 72 °C for 8 min. The corresponding RT-PCR primers were used for chloroplast RNA splicing analysis according to previous studies [18,26].

### 3.6. Chlorophyll Concentration and F_v_/F_m_ Measurement

Total chlorophyll content in the plants was determined spectrophotometrically. In brief, we obtained 0.1 g leaf samples at the three leaves stage, cut them into pieces, and immersed them into 15 mL 95% ethanol for 48 h in darkness at 4 °C. The chlorophyll concentrations were measured using a UV-1800PC (Mapada, China) spectrophotometer at 665 nm, 649 nm, and 470 nm. According to the method of Lichtenthaler [34], we calculated Chla, Chlb, and Car content pigment concentrations. 

In three leaves stage, WT and *oscaf1* mutants were treated with darkness for 30 min, and Fv/Fm was measured using a handheld fluorometer according to the manufacturer’s instructions. There were 5 WT and mutant plants used for analysis, respectively. The values in the figures represent the means ± SE. Statistical significance was analyzed using Student’s *t*-test * and ** indicate *p* < 0.05 and *p* < 0.01, respectively.

### 3.7. Subcellular Localization of OsCAF1

We cloned the coding sequence of *OsCAF1,* and the PCR product was fused to the N-terminus of the green fluorescent protein (GFP) in the pAN580 vector. Both the OsCAF1–GFP fusion construct and the empty GFP vector were transformed into rice protoplasts as previously described [35]. Protoplasts with GFP signals were observed under a laser scanning confocal microscope (Zeiss LSM700, Germany). The PCR amplification primers are listed in Appendix A.

### 3.8. Yeast Two-hybrid Analysis

The coding sequences of *OsCAF1* and *OsCRS2* (*Os01g0132800*) were amplified using gene-specific primers. Full-length *OsCAF1* and partial-length *OsCAF1* were cloned into pGBKT7 (BD) to generate the BD-OsCAF1, BD-OsCAF1-N, BD-OsCAF1-M, BD-OsCAF1-C, BD-OsCAF1-C1, BD-OsCAF1-C2, BD-OsCAF1-C3, and BD-OsCAF1-C4 plasmids. In addition, full-length OsCRS2 was cloned into pGADT7 (AD) to generate the AD-OsCRS2 plasmid. The pairwise plasmid was co-transformed into the AH109 yeast strain and growth on SD-T/L in a 28 °C incubator. Next, the co-transformed yeast strains were transferred to SD-T/L/H/A for interaction analysis.

### 3.9. Measurement of H_2_O_2_ Content

The H_2_O_2_ was extracted using 3-amino-1,2,4-triazole, and according to the method of Brennan and Frenkel [36]. In brief, we obtained 0.1-g leaf samples at the three leaves stage. The samples were homogenized in 3 mL of 3-amino-1,2,4-triazole (10 mM) and centrifugation for 20 min under 6000× *g*. Next, 1mL (0.1% titanium tetrachloride to 20% H_2_SO_4_) was added to 1.5mL supernatant. Centrifuge the reaction solution to remove insoluble materials and measured absorbance at 410 nm against a blank. The standard curve was used to calculate the content of H_2_O_2_.

## Figures and Tables

**Figure 1 ijms-20-04386-f001:**
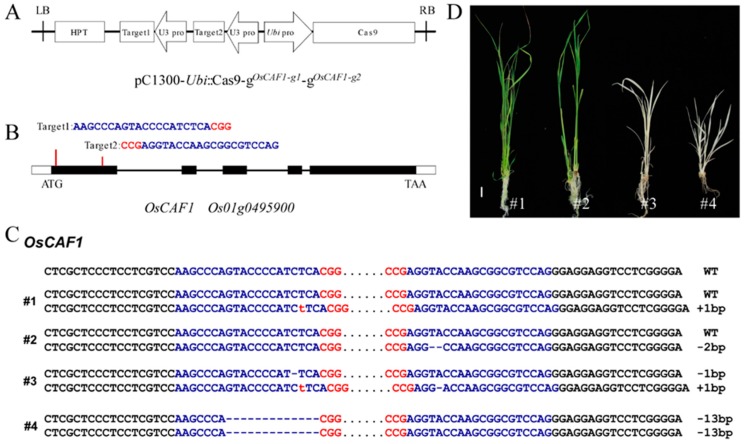
Knockout of *OsCAF1* produces the albino phenotype in seedlings. (**A**) Diagram of CRISPR/Cas9 system for editing OsCAF1. (**B**) Schematic diagram of the targeted site in OsCAF1. (**C**) Mutation types of four positive plants in T_0_ generation. The targeted sequences are in blue letters. The protospacer adjacent motif (PAM) sequences are in red letters. The number of nucleotides at both sides of the target sequences is 18 bp. (**D**) Phenotypes of four positive plants at seedling stage. Scale bar, 1 cm.

**Figure 2 ijms-20-04386-f002:**
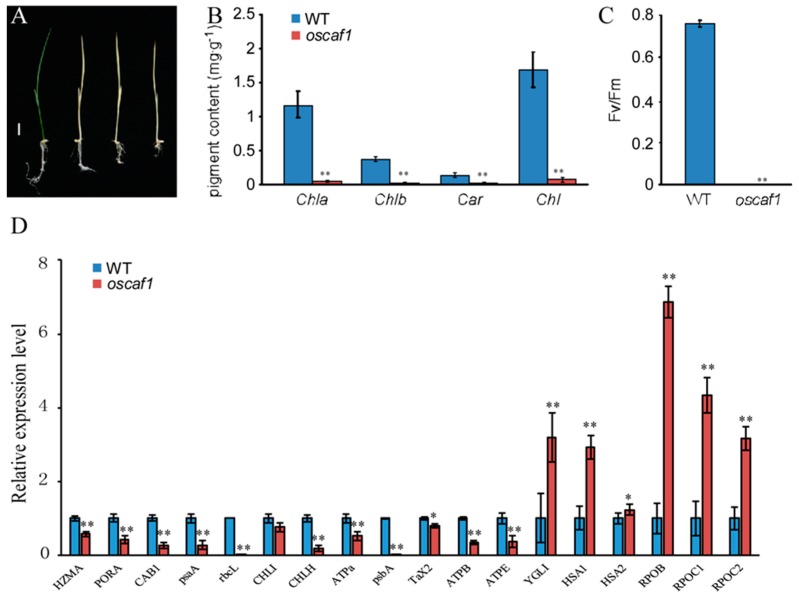
*oscaf1* showed defects in chloroplast development and photosynthesis. (**A**) Phenotype observations of WT (left) and *oscaf1* mutant (right). Scale bar, 1 cm. (**B**) Pigment content of WT and *oscaf1* at seedling stage. (**C**) Photochemical efficiency of PSII in the light (Fv/Fm) of WT and *oscaf1* mutants at seedling stage. (**D**) Relative expression levels of chloroplast development and photosynthesis genes in WT and *oscaf1* mutants. The data represent mean ± SE from three independent biological duplicate, * and ** indicated *p* < 0.05 and *p* < 0.01, respectively.

**Figure 3 ijms-20-04386-f003:**
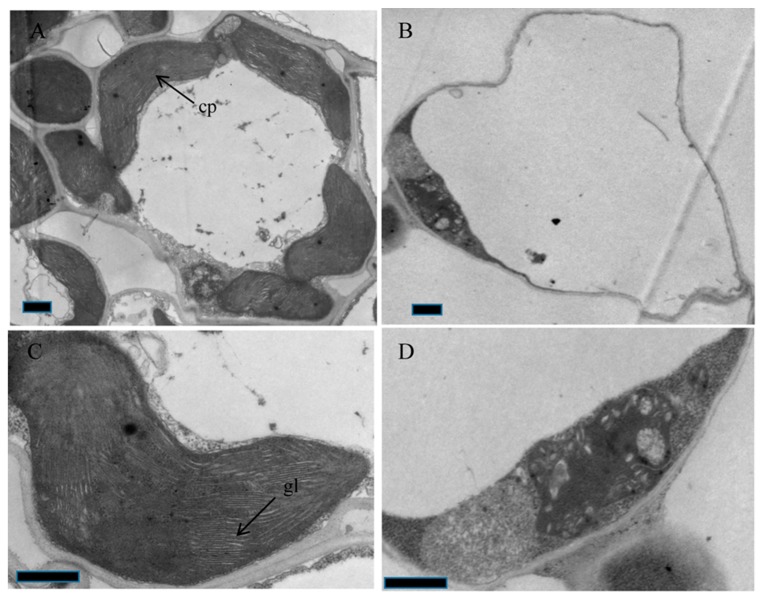
Chloroplast ultrastructure observation of WT (**A**,**C**) and *oscaf1* mutants (**B**,**D**) mesophyll cell at the three leaves stage. cp, chloroplast, gl, grana lamella. Bars = 2.0 μm.

**Figure 4 ijms-20-04386-f004:**
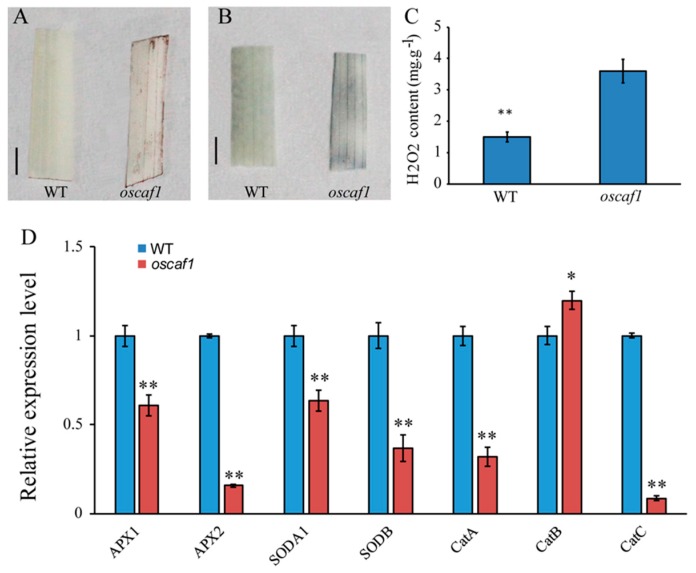
DAB, NBT staining, H_2_O_2_ contents and gene expression level analysis in WT and *oscaf1* mutants. (**A**) DAB staining of leaves from WT (left) and oscaf1 mutants (right). Scale bar, 1 cm. (**B**) NBT staining of leaves from WT (left) and oscaf1 mutants (right). Scale bar, 1 cm. (**C**) Measurement of H_2_O_2_ content in WT and *oscaf1* mutant (**D**) Relative expression levels of genes involved in ROS-scavenging. The data represent mean ± SE from three independent biological duplicate, * and ** indicated *p* < 0.05 and *p* < 0.01, respectively.

**Figure 5 ijms-20-04386-f005:**
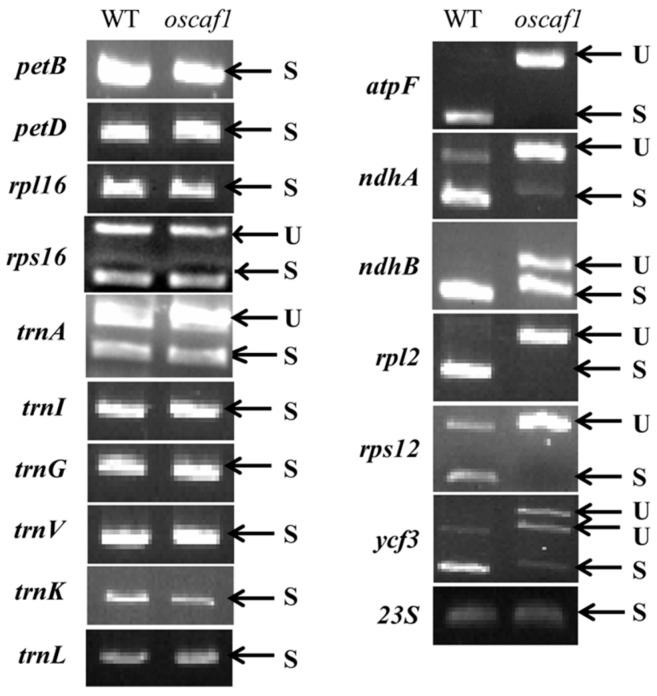
Splicing analysis of chloroplast gene transcripts in WT and *oscaf1* mutants. U, unspliced transcripts; S, spliced transcripts.

**Figure 6 ijms-20-04386-f006:**
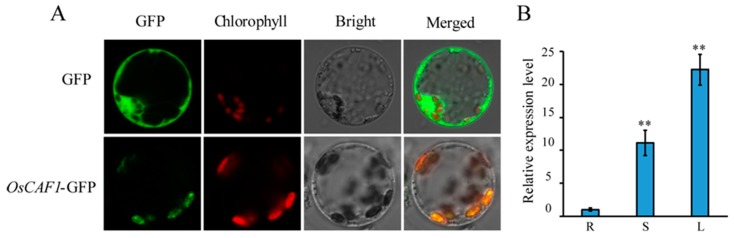
Subcellular localization and expression patterns of OsCAF1. (**A**) Subcellular localization of OsCAF1 in rice protoplasts. (**B**) Expression analysis of OsCAF1 in different tissues. The tissues include roots (R), stems (S), and leaves (L). The values in the figures represent the means ± SE (*n* = 3). ** indicate *p* < 0.01.

**Figure 7 ijms-20-04386-f007:**
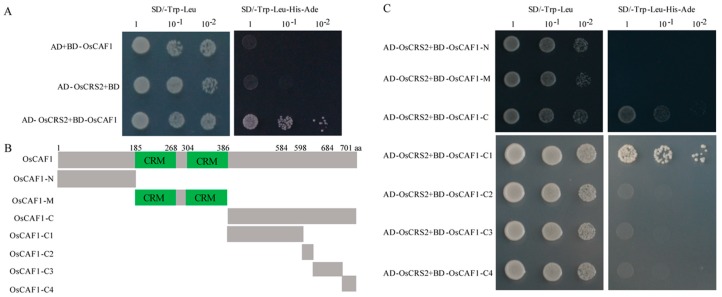
C-terminal of OsCAF1 interacts with OsCRS2. (**A**) OsCAF1 interacts with OsCRS2. The full-length CDS sequences of OsCAF1 and OsCRS2 were cloned into PGBKT7 (BD) and PGADT7 (AD) vector, respectively, to generate BD-OsCAF1 and AD-OsCRS2 constructs. The successfully cloned two constructs were then co-transformed into yeast AH109 cells. AD+ BD-OsCAF1 and BD+AD-OsCRS2 constructs were used as control. (**B**) The protein structure of OsCAF1. Diagram of OsCAF1 protein structure and different structural domains of OsCAF1, OsCAF1-N (1-184), OsCAF1-M (185-386), OsCAF1-C (387-701), OsCAF1-C1 (387-584), OsCAF1-C2 (585-598), OsCAF1-C3 (599-684), and OsCAF1-C4 (685-701). (**C**) Yeast two-hybrid results of the interaction of OsCRS2 and OsCAF1. Truncated versions of OsCAF1 were cloning into BD vector and to generate BD-OsCAF1-N, BD-OsCAF1-M, BD-OsCAF1-C, BD-OsCAF1-C1, BD-OsCAF1-C2, BD-OsCAF1-C3 and BD-OsCAF1-C4 vectors. These vectors were co-transferred to AH109 yeast cells with AD-OsCRS2, respectively. Yeast cells were selected on synthetic dextrose medium lacking Leu and Trp (SD/-Trp-Leu) and the medium lacking Leu, Trp, His, and Ade (SD/-Trp-Leu-His-Ade) with different dilution series (1, 10^−1^, and 10^−2^).

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
