# Peer review of "OsCAF1, a CRM Domain Containing Protein, Influences Chloroplast Development"

_ijms, 2019, doi:10.3390/ijms20184386_

Round 1

Reviewer 1 Report

Qiang et al present a study characterising the rice homologue of CAF1 (OsCAF1) using CRISPR-Cas9 mutation. They perform a series of experiments showing abnormal chloroplast development, increased ROS levels, altered gene expression of both nuclear- and chloroplast-encoded genes and altered splicing of chloroplast-encoded transcripts. Futhermore, they provide evidence of OsCAF1 localisation to the chloroplast and interaction with OsCRS2, providing some mechanistic insight. In general, I think this is a worthwhile study deserving publication, however, I have various concerns which should be met prior.

Major

Could the authors please clarify which caf1 mutant they carried through the paper for analyses and their rationale? Or was it a mix of the CRISPR generated mutants? Is possible, an idea of the the predicted effect on protein structure would be ideal.

How were total carotenoids measured? Could this please be clarified in the methods section - was it based on the Lichtenthaler method?

There is no description of how Fv/Fm was measured - could this please be included in the method section.

I think there needs to be clarification of the authors' conclusion of the Fv/Fm results. While it confirms that the chloroplasts are non-functional, I do not think it suggests an impairment of photosynthesis directly. Instead, the results suggest impaired chloroplast development, which then limits the ability for photosynthesis to occur (fewer functional chloroplasts) rather than affecting photosynthesis directly (e.g. any functional chloroplasts in the caf1 mutant likely can photosynthesis properly). Thus, I think this sentence should just state chloroplast development.Thus, please delete "and photosynthesis" in the paragraph starting L.111. 

There is no description of the statistical analyses performed in this study, although P-values are reported which I assume came from t-tests? For almost all the data (Figs 2D, 4C, 5B), these should be obtained using an ANOVA and post-hoc Tukey comparisons with P-value correction for multiple hypothesis testing. In the case of Fig 2C (Fv/Fm), a simple t-test should suffice. Please include a description of these statistical analyses in the methods.

Could the authors include any details on their experimental design? Which experiments were performed in paddy fields and which in the growth chamber? Was there any randomisation for planting? 

I do not find the ROS assays convincing as they stand, particularly as only 1 replicate is shown (though the authors imply multiple plants were stained) and the picture is not very clear. Based on experience, while these stains can give crude preliminary results - they are highly variable and unreliable. I would point the authors towards Noctor, Mhamdi, Foyer 2016 to pick a more suitable method for assessing ROS levels. While using some of the proposed transgenic methods are exhaustive, in vitro quantification of extracted ROS should be suitable for the purposes here (e.g. Box 1 for H202). I am not sure how much can be concluded from the NBT staining, though there is not a simple method for quantifying singlet oxygen. A quantification of H20to complement DAB staining should suffice as well as considering/presenting replication of both the DAB and NBT stained leaves.

Based on the provided reference [25], trnA and rps16 should have unspliced transcripts in WT in Fig 5? 

The microscopy figures in Fig 6 are quite low resolution, and it makes it hard to have confidence in the localisation of GFP in the OsCAF1-GFP line. This is critical to ensure that we can observe distinct aggregates of GFP within the chloroplast rather than diffuse expression surrounding the chloroplasts. The localization of OsCRS2 should also be performed to give further evidence of an in vivo interaction with CAF1 (ie. does CRS2 co-localize with CAF1) and complementing the Y2H results.

Can the authors please provide more details for the data underpinning Fig 6B - how were the sections made? Are those error bars reflecting biological variance? 

L. 221, though reduced chloroplast numbers can be inferred, proper quantification/counting should be done to make this claim. Similarly, there is no quantification of leaf mesophyll cells.

L. 225, the sentence beginning with "OsCAF1 could play ..." could be clarified and stated more definitely. Early plant growth is evident here, suggesting CAF1 is not necessary for that, but loss of CAF1 is clearly critical for normal plant growth. Could the authors also elaborate on the proposed link to chlorophyll production? To me, the data suggest a lack of chlorophyll as a result of impaired chloroplast development.

Was there any check to ensure in-tact RNA was used for qRT-PCR or alternative splicing analysis (e.g. semi-denaturing agarose gel)? This is critical to ensure differences are biological and not technical. I would also highly recommend not using the 2-ΔΔCT method for quantifying gene expression as it does not take into account PCR efficiency (unless the authors performed a dilution series to check primer efficiencies and incorporated that into the calculation - please explain if so) nor baseline fluorescence. I highly recommend re-analyses of the raw qPCR data using LinReg (Ramakers etal 2002, Ruitjer et al 2009). To clarify, this should not require further qPCR but instead a reanalysis of the existing raw fluorescence values. Alternatively, if a dilution series can be performed on the cDNA generated from a pooled RNA sample for all primers to obtain PCR efficiency values and incorporate these into the calculations performed (include description in methods).

Minor

L.48 - appears to be a speculative sentence and is unclear whether the authors are proposing this as a hypothesis in their study or an observed phenomenon from the literature. Could the authors please clarify. 

The introduction should give some background to the genes undergoing splicing in the chloroplast and why this is important. At least a couple of sentences should be included to facilitate comprehension by non-experts (e.g. for the paragraph beginning L.52).

L31 "organelle" should be "organelles".

L.32 "nucleic" should be "nuclear-encoded"; delete "by".

L.33 "chlorophyll" should be "chloroplasts"

L. 80 "PEP" should be "PEP-dependent genes"

L. 99. Can the authors please clarify why they describe growth conditions both under paddy fields and in control growth chambers when there hasn't really been consideration of the differences between the two. For instance, how were the seedlings in Fig 1D grown?

Fig 2 legend L.127, please specify homozygous caf1 mutant.

Can the authors please clarify "three independent biological duplicate" mentioned multiple times in their legend. Are these three biological replicates assayed in technical duplicate?

L.144, The starting sentence could be stated more definitely as I think this is now well-established. A more appropriate citation would be Noctor, Reichheld, Foyer 2018 Semin Cell Dev Biol.

L. 154, Reduced gene expression suggests lower enzyme levels. To conclude lower activity or concentration, an activity assay or Western blot would need to be performed. Please amend wording. Likely, this reduced expression is feedback from fewer functional chloroplasts in Oscaf1.

Could the authors please be consistent with the writing of group II A vs group IIA (or II B vs IIB).

Fig 6, check formatting of "chlorophyll" label.

It may be worthwhile for the authors to run a motif search (e.g, see tools http://meme-suite.org/) on the various section of OsCRS2 defined in Fig 7, especially of those highly conserved regions. 

Please fix figure legend for Fig 7 (e.g. "(B)" is incorrectly placed and there is no "(C)"). Can the grey bars for 7A also be edited to reflect the defined regions for the C1, C2, C3 and transgenics (e.g. C1 should stop at 584 and C2 should start at 585).

L. 223 delete "reduced".

L. 288 delete "and"; change "photosynthetic process-related" to "photosynthesis-associated".

L. 231 the final sentence is a more likely explanation that influencing chlorophyll production. Is there any literature around how impaired splicing affects PEP activity?

L. 249 "nhdB" should be "ndhB".

L. 256 "lethality" should be "lethal".

L. 356-57 Could the authors please elaborate on the methods for Y2H. The details in the cited paper gives negligible info and, thus, is not an appropriate citation.

L. 363, so are leaves in Fig 4A representative? Could some other replicates be included to give confidence in the ROS staining results.

Author Response

Major

Could the authors please clarify which caf1 mutant they carried through the paper for analyses and their rationale? Or was it a mix of the CRISPR generated mutants? Is possible, an idea of the the predicted effect on protein structure would be ideal.

Response:Thank you for your comments. Both the #1 and #2 OsCAF1/oscaf1 heterozygous plants (Figure 1D) could generate the oscaf1/oscaf1 albino seedlings in T1 generation, and the sequencing results showed that these albino seedlings were caused by the gene disruption of OsCAF1 in extron1. The oscaf1 mutation from plant #1 was one base insertion at the 58th in the CDS region of OsCAF1 gene that leading to the early termination of OsCFA1 translation, while the two base deletion of 537-538th in the CDS region also leading to the early termination of translation. As the same phenotype and function loss of the homozygous mutations, we chose the albino seedlings isolated from plant #1 T1 generation (Figure 2A) as oscaf1 mutant for further study.  

How were total carotenoids measured? Could this please be clarified in the methods section - was it based on the Lichtenthaler method?

Response:Thank you for your good suggestions. The total carotenoids measured method was according to Lichtenthaler method. In this revised, the method of the total carotenoids was added in method section and marked in red.

There is no description of how Fv/Fm was measured - could this please be included in the method section.

Response:Thank you for your comments. In the revised, we added the measure method of Fv/Fm in method section and marked in red.

I think there needs to be clarification of the authors' conclusion of the Fv/Fm results. While it confirms that the chloroplasts are non-functional, I do not think it suggests an impairment of photosynthesis directly. Instead, the results suggest impaired chloroplast development, which then limits the ability for photosynthesis to occur (fewer functional chloroplasts) rather than affecting photosynthesis directly (e.g. any functional chloroplasts in the caf1 mutant likely can photosynthesis properly). Thus, I think this sentence should just state chloroplast development. Thus, please delete "and photosynthesis" in the paragraph starting L.111. 

Response:Thanks for your advice. In this revision, we delete "and photosynthesis".

There is no description of the statistical analyses performed in this study, although P-values are reported which I assume came from t-tests? For almost all the data (Figs 2D, 4C, 5B), these should be obtained using an ANOVA and post-hoc Tukey comparisons with P-value correction for multiple hypothesis testing. In the case of Fig 2C (Fv/Fm), a simple t-test should suffice. Please include a description of these statistical analyses in the methods.

Response:Thanks for your good suggestion. In this revision, we added the statistical analysis method to methods 4.4 and 4.6 and marked in red

Could the authors include any details on their experimental design? Which experiments were performed in paddy fields and which in the growth chamber? Was there any randomisation for planting? 

Response:All of the transgenic T0 generation plants were planted in chamber of greenhouse. The leaves of the heterozygous OsCAF1 /oscaf1 mutants were green (Figure1 D #1 and #2), but the leaves of oscaf1 /oscaf1 mutants showed albinism (Figure1 D #3 and #4). Seeds of heterozygous mutant OsCAF1 were randomly planted in chamber. In T1 generation, the leaves of the heterozygous mutants were green, while the leaves of oscaf1/oscaf1 mutants (allelic variation of OsCFA1) showed albino seedlings. We just showed the phenotype in chamber (Figure 2A), so we removed the “paddy fields” from this version.

I do not find the ROS assays convincing as they stand, particularly as only 1 replicate is shown (though the authors imply multiple plants were stained) and the picture is not very clear. Based on experience, while these stains can give crude preliminary results - they are highly variable and unreliable. I would point the authors towards Noctor, Mhamdi, Foyer 2016 to pick a more suitable method for assessing ROS levels. While using some of the proposed transgenic methods are exhaustive, in vitro quantification of extracted ROS should be suitable for the purposes here (e.g. Box 1 for H202). I am not sure how much can be concluded from the NBT staining, though there is not a simple method for quantifying singlet oxygen. A quantification of H20to complement DAB staining should suffice as well as considering/presenting replication of both the DAB and NBT stained leaves.

Response:Thanks for your suggestion. In the experiment, we stained leaves of multiple WT and mutant leaves with DAB and NBT. The results showed that the leaves of the mutant leaves were staining darker than WT leaves. Figure 4A and Figure 4B are typically shown. We were measurement the content of H2O2 in WT and mutants leaves. In this revision, we add the H2O2 content.

Based on the provided reference [25], trnA and rps16 should have unspliced transcripts in WT in Fig 5? 

Response:The chloroplast gene trnA and rps16 were normal spliced in WT and oscaf1 mutant. In this revision we show trnA and rps16 spliced and unspliced transcripts in Fig 5.

The microscopy figures in Fig 6 are quite low resolution, and it makes it hard to have confidence in the localisation of GFP in the OsCAF1-GFP line. This is critical to ensure that we can observe distinct aggregates of GFP within the chloroplast rather than diffuse expression surrounding the chloroplasts. The localization of OsCRS2 should also be performed to give further evidence of an in vivo interaction with CAF1 (ie. does CRS2 co-localize with CAF1) and complementing the Y2H results.

Response:Thank you for your comments. In this revision, we provide high-resolution subcellular localization in Figure 6 of OsCAF1-GFP line. The location of OsCRS2 subcellular in chloroplasts was predicted by using TargetP (http://www.cbs.dtu.dk/services/TargetP/). In addition, OsCRS2 could interaction with OsCAF1 that subcellular in chloroplasts. The function of OsCRS2 is being studied in detail; the subcellular location of OsCRS2 is not shown in this study. We'll show that in next studies.

Can the authors please provide more details for the data underpinning Fig 6B - how were the sections made? Are those error bars reflecting biological variance? 

Response:Thank you for your comments. In three leaves stages, we obtain roots, stems and leaves in WT. The values in the figures represent the means ± SE. The error bars reflecting SE.

221, though reduced chloroplast numbers can be inferred, proper quantification/counting should be done to make this claim. Similarly, there is no quantification of leaf mesophyll cells.

Response:Thank you for your comments. In this revision, we changed the original to “In plant leaves, the chloroplast numbers in oscaf1 mutants decreased, accompanied by abnormal chloroplast development (supplementary figure 1) and damaged thylakoid membranes compared with the WT, which led to decrease in chlorophyll contents in oscaf1 mutants.”  In addition, we counted the number of chloroplasts in the cells and put this in supplementary figure 3.

225, the sentence beginning with "OsCAF1 could play ..." could be clarified and stated more definitely. Early plant growth is evident here, suggesting CAF1 is not necessary for that, but loss of CAF1 is clearly critical for normal plant growth. Could the authors also elaborate on the proposed link to chlorophyll production? To me, the data suggest a lack of chlorophyll as a result of impaired chloroplast development.

Response: Thanks for your suggestion. In this revision, we changed this sentence “OsCAF1 could play a key role in the early development of chlorophyll and could be essential for normal plant growth” with “OsCAF1might play an important role in chloroplast development”

Was there any check to ensure intact RNA was used for qRT-PCR or alternative splicing analysis (e.g. semi-denaturing agarose gel)? This is critical to ensure differences are biological and not technical. I would also highly recommend not using the 2-ΔΔCT method for quantifying gene expression as it does not take into account PCR efficiency (unless the authors performed a dilution series to check primer efficiencies and incorporated that into the calculation - please explain if so) nor baseline fluorescence. I highly recommend re-analyses of the raw qPCR data using LinReg (Ramakers etal 2002, Ruitjer et al 2009). To clarify, this should not require further qPCR but instead a reanalysis of the existing raw fluorescence values. Alternatively, if a dilution series can be performed on the cDNA generated from a pooled RNA sample for all primers to obtain PCR efficiency values and incorporate these into the calculations performed (include description in methods).

Response: Thanks for your suggestion. We tested RNA integrity with semi-denaturing agarose gel before RT-PCR and qRT-PCR. The results showed that the RNA of the WT and mutant was intact. We are re-analyses of the raw qPCR data using LinReg method (Peirson et al.2003, Ruijter et al.2009) and cite two reference in this revision.

Minor

L.48 - appears to be a speculative sentence and is unclear whether the authors are proposing this as a hypothesis in their study or an observed phenomenon from the literature. Could the authors please clarify.

Response: We observed abnormal chloroplast development phenomenon from related mutants according to literature reported. In this revision, we were addition the cited references.

 The introduction should give some background to the genes undergoing splicing in the chloroplast and why this is important. At least a couple of sentences should be included to facilitate comprehension by non-experts (e.g. for the paragraph beginning L.52).

Response: Thanks for your suggestion. In this revision, we added two sentences beginning L52. “In plants, splicing of chloroplast introns requires splicing factors encoded by nuclear genes [8]. And abnormal splicing of chloroplast intron will affect the development of chloroplast [14-16].”

L31 "organelle" should be "organelles".

Response: Thanks for your suggestion. In this revision, we are removing “organelle” and writer “organelles” 

L.32 "nucleic" should be "nuclear-encoded"; delete "by".

Response: Thanks for your suggestion. In this revision, we delete "by" and removing "nucleic" and writer "nuclear-encoded".

L.33 "chlorophyll" should be "chloroplasts".

Response: Thanks for your suggestion. In this revision, we are removing "chlorophyll" and writer "chloroplasts ".

80 "PEP" should be "PEP-dependent genes".

Response: Thanks for your suggestion. In this revision, we removing " PEP " and writer " PEP-dependent genes ".

99. Can the authors please clarify why they describe growth conditions both under paddy fields and in control growth chambers when there hasn't really been consideration of the differences between the two. For instance, how were the seedlings in Fig 1D grown?

Response: Thanks for your suggestion. In the field and chambers planting, the different OsCFA1 allelic variation mutants were showed albino seedlings phenotypic. The OsCAF1/oscaf1 heterozygotes plants are showed normal green leaves phenotypes. The Fig 1D showed plant were grown under chambers. We didn't show the phenotype in the paddy fields. So, in this revision, we delete “paddy fields”.

Fig 2 legend L.127, please specify homozygous caf1 mutant.

Response: We were chose the albino seedlings isolated from plant #1 T1 generation as oscaf1 (Figure 2A) for further study. In this revision, we explained the origin of oscaf1 mutant in detail.

Can the authors please clarify "three independent biological duplicate" mentioned multiple times in their legend. Are these three biological replicates assayed in technical duplicate?

Response: Each biological duplicate contains three technical duplicate.

L.144, The starting sentence could be stated more definitely as I think this is now well-established. A more appropriate citation would be Noctor, Reichheld, Foyer 2018 Semin Cell Dev Biol.

Response: Thanks for your suggestions. In this revision, we were citation (Noctor et al, 2018) reference.

154, Reduced gene expression suggests lower enzyme levels. To conclude lower activity or concentration, an activity assay or Western blot would need to be performed. Please amend wording. Likely, this reduced expression is feedback from fewer functional chloroplasts in Oscaf1.

Response: Thanks for your suggestions. In this revision, we are re-writing this sentence.

Could the authors please be consistent with the writing of group II A vs group IIA (or II B vs IIB).

Response: Thanks for your suggestions. In this revision, we writing of group II A vs group IIA and group II B vs group IIB.

Fig 6, check formatting of "chlorophyll" label.

Response: Thanks for your suggestions. In this revision, we rewrite the "chlorophyll" label.

It may be worthwhile for the authors to run a motif search (e.g, see tools http://meme-suite.org/) on the various section of OsCRS2 defined in Fig 7, especially of those highly conserved regions.

 Response: Thanks for your suggestions. Through sequence alignment revealed that the highly conserved domain of OsCRS2, containing peptidyl-trna hydrolase that required for the splicing of group II introns in chloroplasts. This conserved domain contents has analysis.

Please fix figure legend for Fig 7 (e.g. "(B)" is incorrectly placed and there is no "(C)"). Can the grey bars for 7A also be edited to reflect the defined regions for the C1, C2, C3 and transgenics (e.g. C1 should stop at 584 and C2 should start at 585).

Response: Thanks for your suggestions. In this revision, we re-writing Fig 7 figure legend.

223 delete "reduced".

Response: Thanks for your suggestions. In this revision, we delete "reduced".

288 delete "and"; change "photosynthetic process-related" to "photosynthesis-associated".

Response: Thanks for your suggestions. In this revision, we change "photosynthetic process-related" to "photosynthesis-associated".

231 the final sentence is a more likely explanation that influencing chlorophyll production. Is there any literature around how impaired splicing affects PEP activity?

Response: Some studies (Such as reference [23, Albino Leaf 2 , 33 WSL5 ] ) have reported that the impaired splicing of chloroplast genes might affect the expression levels of PEP-related genes, but how to affect the PEP activity is rarely reported.

249 "nhdB" should be "ndhB".

Response: Thanks for your suggestions. In this revision, we change " nhdB " to " ndhB ".

256 "lethality" should be "lethal".

Response: Thanks for your suggestions. In this revision, we change "lethality " to " lethal ".

356-57 Could the authors please elaborate on the methods for Y2H. The details in the cited paper gives negligible info and, thus, is not an appropriate citation.

Response: Thanks for your suggestions. In this revision, we were introduced the method of Y2H in detail and remove this reference citation.

363, so are leaves in Fig 4A representative? Could some other replicates be included to give confidence in the ROS staining results.

Response: We staining multiple leaves of oscaf1 mutant, and Figure 4A were a representative. In this review, we add H2O2 content in oscaf1 mutant leaves.

Reviewer 2 Report

To know functions of CRM domain containing proteins in chloroplasts developmet, authors have chosen an OsCAF1 protein as a target. They made knockout mutants of OsCAF1 using a CRISPR-Cas9 system and found that those mutants shows albino phenotype. They also identified that knockout of OsCAF1 leads splicing defects of some chloroplast genes. They concluded that functions of OsCAF1 are different from those of AtCAF1 and ZmCAF1.

              This conclusion is very interesting, however, there are many problems in this manuscripts. Although they made four allelic mutants, the results obtained from only one strains, oscaf1. Which allele did you use as an oscaf1 mutant? There are no biological replications. Also, which kinds of change are led by those mutations? They investigated localization of OsCAF1 protein using OSCAF1-GFP expression vector. In a supplementary list, I could find primers for a vector construction. The revers primer, P580-OsCAF1-R contains stop codon. I think that this OsCAF1-GFP construct is unfunctional. Are the GFP signals in the Figure 6A obtained from OsCAF1-GFP?

Minor points

There are many typos in this ms. Please consult professional English editing service or native speaker.

Author Response

This conclusion is very interesting, however, there are many problems in this manuscripts. Although they made four allelic mutants, the results obtained from only one strains, oscaf1. Which allele did you use as an oscaf1 mutant? There are no biological replications. Also, which kinds of change are led by those mutations? They investigated localization of OsCAF1 protein using OSCAF1-GFP expression vector. In a supplementary list, I could find primers for a vector construction. The revers primer, P580-OsCAF1-R contains stop codon. I think that this OsCAF1-GFP construct is unfunctional. Are the GFP signals in the Figure 6A obtained from OsCAF1-GFP?

Response:  In this study, the #1 and #2 OsCAF1/oscaf1 heterozygous plants (Figure 1D) could generate the albino seedlings in T1 generation. Sequencing results showed that the oscaf1 mutant from #1 in T1 generation were cause by one base insertion at the 58th in the CDS region of OsCAF1, leading to the formation of a premature stop codon. The other OsCFA1 allelic mutant from #2 in T1 generation also showed albino seedlings phenotype, and the mutation was the deletion of two base at position 537 and 538 in coding region of OsCAF1, caused the reading frame shift of OsCAF1, leading to the formation of a premature stop codon. Similarly, the albino seedlings #3 and #4 plants in the T0 generation caused by the mutation of lack of bases in CDS of OsCAF1. The deletion mutation induced the reading frame shift of OsCAF1, leading to the formation of a premature stop codon. Above all, these four allelic mutants caused the reading frame shift of OsCAF1, leading to the formation of a premature stop codon and eventually resulted in the albino seedlings phenotype. So, we choose the oscaf1 mutant from plant #1 in T1 generation for further research.

    We appreciate your suggestion. We re-check the constructed OsCAF1-GFP vector sequence and the amplified primer. We find the OsCAF1-GFP vector do not content the stop codon of OsCAF1 (see as Figure 1 in below). So this OsCAF1-GFP construct is functional. In order to obtain clearer GFP signals, we re-transferred OsCAF1-GFP vector into rice protoplasts. In this revision, we replaced the Figure 6A with new figure. We also provided the correct primer (P580-OsCAF1-F: GACAGCCCAGATCAACTAGTATGGCAACTAGCCACCTCACCTC ; P580-OsCAF1-R : CCCTTGCTCACCATGGATCCAGCCAGTAATTTAGCAAGTTCATCTAC) for OsCAF1-GFP vector construction in the supplementary files. We apologized for not checking the primer sequence (see figure in the attached file)

Minor points

There are many typos in this ms. Please consult professional English editing service or native speaker.

Response: Thanks for your reminding. We checked spelling and consulted with a professional English editor to polish the manuscript.

Round 2

Reviewer 1 Report

The authors have extensively addressed all of my concerns. There is still a need for minor proof-reading to fix some typos and grammatical errors.

Author Response

The authors have extensively addressed all of my concerns. There is still a need for minor proof-reading to fix some typos and grammatical errors.

Respond:Thank you for your comments. In this revision, we carefully check the spelling and grammar of the manuscript and revise it.

Reviewer 2 Report

Line 100-102, “Both 3 and #4 mutants cause by …” Meaning is unclear. Do you mean that “In both of the #3 and #4 mutants, a premature stop codon was created by a frame shift in the coding region of OsCAF1.”?

Could you remove the sentence Line 106-108, “ Homozygous oscaf1/ocaf1 ….”, to avoid a redundancy.

Meaning of the sentence, line 164-165, is unclear. Are there functional chloroplasts in the oscaf1 mutant?

Author Response

Line 100-102, “Both 3 and #4 mutants cause by …” Meaning is unclear. Do you mean that “In both of the #3 and #4 mutants, a premature stop codon was created by a frame shift in the coding region of OsCAF1.”?

Response: Thanks for your comment. “Both 3 and #4 mutants cause by …”that “In both of the #3 and #4 mutants, a premature stop codon was created by a frame shift in the coding region of OsCAF1.”? In this revision, we have modified this sentence.

Could you remove the sentence Line 106-108, “ Homozygous oscaf1/ocaf1 ….”, to avoid a redundancy.

Response: Thanks for your comment. In this revision, we remove the sentence Line 106-108, “ Homozygous oscaf1/ocaf1 ….”

Meaning of the sentence, line 164-165, is unclear. Are there functional chloroplasts in the oscaf1 mutant?

Response: Thanks for your comment. In this revision, we have modified this sentence to “The ROS-scavenging genes reduced expression in oscaf1 might feedback from lack of OsCAF1”